# High efficient *de novo* root-to-shoot organogenesis in *Citrus jambhiri* Lush.: Gene expression, genetic stability and virus indexing

Tongbram Roshni Devi[1☉], Madhumita Dasgupta[1☉], Manas Ranjan Sahoo[1]*, Paresh Chandra Kole[2], Narendra Prakash[1]

**1** ICAR Research Complex for North Eastern Hill Region, Imphal, Manipur, India, **2** Institute of Agriculture, Visva-Bharati, Sriniketan, West Bengal, India

☉ These authors contributed equally to this work.
* manas.sahoo@icar.gov.in

**Data Availability Statement:** The minimal anonymized data set has been uploaded to a public repository and is accessible via the following DOIs:

## Abstract

A protocol for high-frequency direct organogenesis from root explants of Kachai lemon (*Citrus jambhiri* Lush.) was developed. Full-length roots (~3 cm) were isolated from the *in vitro* grown seedlings and cultured on Murashige and Skoog basal medium supplemented with Nitsch vitamin (MSN) with different concentrations of cytokinin [6-benzylaminopurine, (BAP)] and gibberellic acid (GA$_3$). The frequency of multiple shoot proliferation was very high, with an average of 34.3 shoots per root explant when inoculated on the MSN medium supplemented with BAP (1.0 mg L$^{-1}$) and GA$_3$ (1.0 mg L$^{-1}$). Optimal rooting was induced in the plantlets under half strength MSN medium supplemented with indole-3-acetic acid (IAA, 0.5–1.0 mg L$^{-1}$). IAA induced better root structure than 1-naphthaleneacetic acid (NAA), which was evident from the scanning electron microscopy (SEM). The expressions of growth regulating factor genes (*GRF1* and *GRF5*) and GA$_3$ signaling genes (*GA2OX1* and *KO1*) were elevated in the regenerants obtained from MSN+BAP (1.0 mg L$^{-1}$)+GA$_3$ (1.0 mg L$^{-1}$). The expressions of auxin regulating genes were high in roots obtained in ½ MSN+IAA 1.0 mg L$^{-1}$. Furthermore, indexing of the regenerants confirmed that there was no amplicons detected for Huanglongbing bacterium and *Citrus tristeza* virus. Random amplified polymorphic DNA (RAPD) and inter simple sequence repeat (ISSR) markers detected no polymorphic bands amongst the regenerated plants. This is the first report that describes direct organogenesis from the root explant of *Citrus jambhiri* Lush. The high-frequency direct regeneration protocol in the present study provides an enormous significance in *Citrus* organogenesis, its commercial cultivation and genetic conservation.

## Introduction

Kachai lemon (*Citrus jambhiri* Lush.), indigenous to Kachai village of Manipur, India, is one of the important commercial species among the *Citrus* realm. Conventionally, it is used as the

10.17605/OSF.IO/F9U35 and 10.17605/OSF.IO/P7SFZ.

**Funding:** Department of Biotechnology (DBT), Govt. of India, under the DBT twinning project 'In vitro mass-multiplication and conservation of some endangered Citrus species of NEH Region of India' (BT/PR16132/NER/95/160/2015).

**Competing interests:** The authors have declared that no competing interests exist.

most reliable rootstock source for *Citrus* propagation in south Asia [1] due to its tolerance to *Citrus tristeza* virus. On the other hand, nucellar polyembryony, sexual incompatibility, heterozygosity, and long juvenility are the major limitations for conventional breeding and commercial cultivation of *Citrus* species [2]. Due to polyembryony in nature, it gives rise to varied vigorous nucellar seedlings, which are nearly indistinguishable from zygotic seedlings [3]. Differentiation of nucellar seedlings from zygotic seedlings is practically not feasible *in vivo*, which leads to genetic erosion in *Citrus*. Although vegetative propagation has been progressed to overcome these problems to a certain extent, classical methods of budding and grafting to maintain 'true-to-type' are inherently associated with clonal degeneration, mainly due to viral diseases. The long-term clonal propagation using axillary vegetative parts leads to somaclonal variations or genetic instability, disease susceptibility and virus complexity [4]. Thus, a highly efficient *in vitro* protocol using suitable explants with minimum subculture needs to be established to overcome these issues.

The *in vitro* technique offers an efficient and meaningful tool to derive large scale elite planting materials, genetic conservation, and improvement of *Citrus*. *In vitro* regeneration in *C. jambhiri* was successfully demonstrated deploying wide range of tissues and organs such as shoot tips [5, 6], nodal explants [1, 3, 5], epicotyl segments [7, 8], and cotyledons [9, 10]. However, low multiplication rate and time-consuming culture processes are still prevalent problems in *Citrus* micropropagation [11]. Moreover, explants obtained from the matured plants leads to contamination and poor shoot and root growth *in vitro* [12]. Thus, commercial micropropagation protocol emphasized axillary explants selection from *in vitro* grown seedlings to minimize contamination and obtain 'true-to-type' plants [13]. Moreover, direct shoot organogenesis provides lower somaclonal variations among the regenerants than indirect regeneration via intermediary step of callusing [1]. Among the explants, rootstocks and root explants in *Citrus* better adapt to biotic stresses, particularly to virus complex. Although few reports are available on shoot induction from root segments in *Citrus* [9, 14], the potential of intact root explants for direct organogenesis of *Citrus jambhiri* has not yet been reported.

Growth medium and vitamin supplements play a vital role in *in vitro* morphogenesis. MS medium [15] is one of the most preferred medium for *Citrus* micropropagation [10]. Use of Nitsch vitamins [16] in tissue culture of *Citrus* species is limited. Nitsch vitamins are comprised of folic acid (0.5 mg L$^{-1}$) and d-biotin (0.05 mg L$^{-1}$) along with nicotinic acid (5 mg L$^{-1}$), pyridoxine HCl (0.5 mg L$^{-1}$), thiamine HCl (0.5 mg L$^{-1}$), and glycine (2 mg L$^{-1}$), which help in plant organogenesis in combination with various cytokinins and gibberellins. Moreover, plant growth regulators (PGRs) and vitamins in the basal medium play an important role in the onset of dormancy and determination of shoot and root architecture [17]. Auxins such as NAA, IAA, and indole-3-butyric acid (IBA) play an essential role in root modulation for better adaptation [18] of *in vitro* plants. The directional mobility of cytokinins and auxins in plants regulates several biological pathways and regulatory networks [17].

Cryptic genetic defects owing to somaclonal variation restrict the utility of the *in vitro* system and necessitate a mandatory assessment of the genetic stability of *in vitro* plantlets [2]. Assessment of genetic fidelity of the regenerants is an essential process to examine genetic identity *in vitro*. Molecular tools such as random amplified polymorphic DNA (RAPD) [2, 19], and inter simple sequence repeat (ISSR) [19, 20] are the reliable DNA based markers used to assess the genetic stability in *Citrus*. A bacterium, *Candidatus* Liberabacter asiaticus (CLa/Las) associated with citrus greening (Huanglongbing, HLB) and *Citrus tristeza* virus (CTV) are among the most prevalent pathogens affecting global citrus industry [21]. Therefore, indexing of the regenerants is an essential routine activity in plant tissue culture to assess the plantlet health [22] before hardening.

Growth regulating factors (GRF) are a class of transcription factors involved in regulating cell expansion and proliferation in gibberellin (GA)-mediated response [23]. Generally, GRF genes express in the growing tissue and act as transcription activators [24]. GA response is closely associated with $GA_2$ oxidase 1 (*GA2OX1*), which converts active GA and its precursors into irreversible inactive GA [25] and *ent*-kaurene oxidase 1 (*KO1*). During organogenesis and root initiation, the growth response is profoundly regulated by auxin regulating genes comprising three families, *i.e.*, Aux/IAA, SAURs, and GH3s [26]. Auxin response factors (*ARF1* and *ARF8*) belong to a family of transcription factors that regulate expressions of auxin response genes [27]. PIN-formed (*PIN1* and *PIN5*) proteins are represented by auxin efflux carrier proteins, having a role in auxin transportation [28]. Expression analysis of *GRFs* and auxin regulating genes provides a better understanding of plant growth and development [27]. Root architect regulates water and nutrient uptake in plants, which is influenced by various regulatory factors. Determining root microstructure using a scanning electron microscope (SEM) is an advanced tool over classical anatomical studies [29].

The objective of the present study was to derive a high efficient protocol for the micropropagation of *C. jambhiri* through root explant, assessing the root microstructure, gene expression, genetic fidelity and virus indexing. The present study reports the first attempt to derive a high-throughput direct organogenesis protocol from *in vitro* intact root explants using MS media supplemented with Nitsch vitamins (MSN).

## Materials and methods

### Plant materials and explant preparation

The mature and healthy fruits of *Citrus jambhiri* Lush. were collected from the Kachai village of Ukhrul District, Manipur, India located at 25°14′ N and 94°16′ E and 1662 m above mean sea level. Seeds were removed from the pulp, allowed to dry at room temperature for 2–3 days (d). Such dry seeds were soaked in distilled water for 20–30 min, and testa was removed by hand without any bruises on the thin inner layer of tegmen. After peeling off the hard outer seed coat testa, seeds were surface sterilized in 2% sodium hypochlorite solution (made from a 4% (w/v) NaClO solution; HiMedia®, Mumbai, India) with 2–3 drops of 2% Tween-20 by vigorous shaking for 5 min. The seeds were then rinsed with sterile distilled water three times followed by washing with 70% ethanol for 30 s. The surface-sterilized seeds were allowed to dry completely inside the laminar airflow (LABTOP, Labtop Instruments Pvt. Ltd, India) and after that soaked in sterile distilled water overnight. The tegmen was removed carefully by using sterile forceps and scalpel. The white seeds portions, stripped of both the seed coats, were inoculated in 25x100 mL test tubes (Borosil®, Mumbai, India) containing 25 mL of culture medium.

### Culture media and conditions

For germination of seeds, half-strength MS [15] basal medium (1-L powder sachet from HiMedia®), supplemented with 3% sucrose, was used. The pH of the culture medium was adjusted to 5.8 with 1 N NaOH, and 0.3% (w/v) of clarigel™ (HiMedia®, India) was added as a gelling agent. The medium was sterilized in an autoclave (Equitron Medica Pvt. Ltd, Mumbai, India) at 121°C, under 15 psi for 20 min, allowed to cool up to 55°C. Before pouring the media, filter-sterilized PGRs were added, poured either in test tubes and in planton boxes (7.5x7.5x10 cm, Tarsons, Kolkata, India) and allowed to solidify. The seeds were inoculated vertically half-submerged into the medium. All the cultures were maintained at 24±2°C, with a photoperiod of 16 h, under fluorescent light (Philips, New Delhi, India) with a light intensity of 40 μ mol m$^{-2}$ s$^{-1}$.

## Direct multiple shoot regeneration from *in vitro* raised roots

Full-grown roots, excised from 4–5 weeks (wks) old *in vitro* grown seedlings, were employed for direct regeneration of multiple shoots. MS medium with Nitsch vitamins (MSN) [16] fortified with 1.0 mg L$^{-1}$ each of BAP, kinetin (KIN), adenine sulphate (ADS) and zeatin (ZEA) [HiMedia®, India] was used either alone or in combination with 1.0 or 2.0 mg L$^{-1}$ of GA$_3$ (HiMedia®, India). Data on the percent of explant response, days to shoot initiation, number of shoots per explant, shoot length (cm), and number of leaves per explant were recorded at an interval of 2 wks after shoot initiation.

## Induction of roots

To induce roots, the regenerants were cultured in various concentrations (0.5, 1.0, 1.5 and 2.0 mg L$^{-1}$) of auxins such as NAA and IAA [HiMedia®, India] incorporated in both half (½MSN) and full-strength MS medium supplemented with Nitsch vitamins (MSN). Observations on days to root initiation, number, and length of roots (cm) per explant and the percent of rooting response were recorded after every 2 wks of culture initiation till 12 wks.

## Gene expression analyses

To understand the role of various growth regulating factors (GRF) and GA$_3$ pathway genes during different stages of direct multiple shoot organogenesis from intact *in vitro* roots, quantitative real-time PCR (qRT-PCR) was performed. Total RNA was isolated from *in vitro* roots and leaves under different treatments and time points using RNeasy Plant Mini Kit (QIAGEN, India (P) Ltd.) following the manufacturer's protocol. The sampling was performed at four different growth stages (Stage I to IV). The stage I represented roots without any response, stage II was comprised of nodulated roots, followed by stage III, where shootlets appeared. The stage IV consisted of fully grown plantlets in which young leaves have been sampled for qRT-PCR. The quality of the isolated RNA was quantified using QIAexpert (QIAGEN Hilden, Germany) and agarose gel electrophoresis for single-stranded cDNA synthesis. One microgram of the isolated RNA was converted into cDNA by dual step RT-PCR kit (GCC Biotech (I) Pvt. Ltd., Kolkata, India) and proceeded for targeted gene expression analysis by using Rotor-Gene Q (QIAGEN Hilden, Germany). The expressions of two growth regulating factors, *viz.*, *GRF1* and *GRF5*, and two GA$_3$ pathway genes (*GA2OX1* and *KO1*), were studied (S1 Table). In a similar way, the expressions of auxin regulating genes, *viz.*, *ARF1*, *ARF8*, *PIN1*, *PIN5*, *GH3* and *IAA4* (S1 Table), were analysed from *in vitro* roots obtained in various auxin supplemented media. The qRT-PCR was performed in a 20 μL reaction volume using 10 μL of QuantiNova® SYBR®green PCR master mix (QIAGEN, India), 1μL each of forward and reverse primers, 1 μL of cDNA and 7 μL of RNAse free water. The reaction was started by an incubation of 30 s at 60˚C, followed by 40 cycles of 95˚C for 30 s, 55˚C for 30 s, 72˚C for 30 s and 65-99˚C for 30 s for melting curve as described by Liu et al. (2017) [30]. Each reaction was carried out in duplicate, and relative gene expression was calculated using 2-$^{\Delta\Delta Ct}$ method [31] with *Citrus actin* gene [32] as an endogenous control.

## Scanning electron microscopy (SEM)

The microstructures of 8 wks old roots were studied using SEM imaging. Roots, from the control along with various auxin supplemented media, were viewed and photographed on SEM (Model: JOEL-JSM 5600) using an automated sputter coater (Model: JEOL, JFC-1600) for 3 min at required magnifications from 500x to 2000x as per the standard procedures [33] at

RUSKA Lab, College of Veterinary Science, P.V. Narasimha Rao Telangana Veterinary University (PVNR TVU), Rajendranagar, Hyderabad, India.

## Indexing for HLB and CTV

Screening of two major *Citrus* infecting diseases, Huanglongbing (HLB, *C*La bacterium) and *Citrus tristeza* virus (CTV) in particular, was carried out by reverse transcriptase PCR (RT-PCR) among the regenerants. For HLB detection, the genomic DNA (gDNA) was isolated from the midrib of the leaf tissues using the DNeasy Plant Mini Kit (QIAGEN, India (P) Ltd.) by following the manufacturer's protocol. The primers, *Las* F (5′ GGAGAGGGTGAGTGG AATTCCGA 3′) and *Las* R (5′ ACCCAACATCTAGGTAAAAACC 3′), were designed from the Las 16S rDNA [34]. The PCR was carried out using Go Taq® G2 green master mix (Promega Corporation, US) under the conditions of 94°C for 4 min, followed by 30 cycles of denaturation at 94°C for 45 s, annealing at 56°C for 45 s, extension at 72°C for 1 min, and final extension at 72°C for 10 min in SimpliAmp thermal cycler (Applied Biosystems, Life Technologies®).

For detection of CTV, total RNA was isolated from the leaf midrib of *in vitro* raised plantlets using RNeasy Plant Mini Kit (QIAGEN, India (P) Ltd.) by following the manufacturer's protocol and converted into cDNA by Verso cDNA synthesis kit (Thermo Scientific™, USA). The primers, *K607* F (5′ACTCRCCGTTTGACTCTGTTTAAA3′) and *K608* R (5′GTACCGA ACATATAACTCCAGT3′), were designed based on the viral coat protein according to Sharma et al. (unpublished data) and were procured from GCC Biotech (I) Pvt. Ltd., Kolkata, India. The PCR was carried out under the conditions of initial denaturation at 94°C for 2 min, 35 cycles of 94°C for 30 s, 59°C for 45 s followed by 72°C for 45 s, and 72°C for 10 min of final extension. For the detection of amplicons, PCR products were run on 1.8% agarose gel, and pictures were taken on the E-Box gel documentation imaging system (Vilber Lourmat, France).

## Assessment of genetic fidelity

To assess the genetic stability of *in vitro* raised plantlets, clonal fidelity test was carried out by using PCR based randomly amplified polymorphic DNA (RAPD) and inter simple sequence repeat (ISSR). Total gDNA was isolated from leaf tissues of the *in vitro* raised plantlets under different PGR treatments using DNeasy Plant Mini Kit (QIAGEN, India (P) Ltd.) using QIAcube by following manufacturer's protocol. The isolated gDNA was quantified using a QIAexpert (QIAGEN Hilden, Germany), checked on 0.8% agarose gel electrophoresis, and finally adjusted to a concentration of 50 ng μL$^{-1}$ for enabling PCR.

**RAPD.** Eight RAPD primers (S2 Table) were used to screen the genetic fidelity of the *in vitro* raised plants. PCR was performed in a total volume of 25 μL comprising 50 ng template DNA, 10 pM primer, 12.5 μL of 2X Taq PCR master mix (QIAGEN, India (P) Ltd.). PCR amplification was carried out at 94°C for 4 min, followed by 40 cycles of denaturation at 94°C for 1 min, annealing at 32-34°C (varies with primer) for 1 min, extension at 72°C for 2 min and final extension of 7 min at 72°C in a thermocycler [19].

**ISSR.** To assess the clonal fidelity of the *in vitro* regenerants, eight ISSR primers (S2 Table) were used. PCR amplification was carried out using 50 ng of gDNA, 1 μL of 10 pM primer, 12.5 μL of 2X Taq PCR master mix for a total reaction volume of 25 μl in a thermocycler with reaction conditions by following the method of Rohini et al. (2015) [19]. The annealing temperature was optimized at 42-54°C by gradient PCR for each primer separately.

All the primers (S1 and S2 Tables) were procured from Bioserve Biotechnologies (India) Pvt. Ltd., Hyderabad, India. Genetic integrity, as revealed by the banding patterns of the PCR

products, was checked on 1.8% agarose gel electrophoresis, and pictures were taken on E-Box gel documentation imaging system (Vilber Lourmat, France).

## Acclimatization and hardening

*In vitro* plantlets with well-developed roots were washed thoroughly with sterile distilled water and acclimatized in the mixture of sterile garden soil, sand, and vermicompost in 1:1:1 ratio. The plantlets were irrigated with Hoagland's solution (HiMedia®) [35] twice daily for the next 2 wks. For hardening, the plantlets were maintained at a well-aerated mist house with temperature 26±2˚C. Well established, virus-free and true-to-type micro propagated plants of *C. jambhiri* were then maintained in the polyhouse.

## Statistical analysis

The experiments were laid out in a completely randomized design with five replications under each treatment. The experiments were repeated twice, and the data represented were mean of two experiments with five replications. Data were analysed with suitable transformation wherever it was so required, using analysis of variance (ANOVA), and the differences among the mean values were compared using *Tukey's* test [36]. The statistical analyses were performed using XLSTAT statistical software (XLSTAT Premium 2020.2.1, Adinsoft, NY).

## Results

### Multiple shoot regeneration from root explants

Depiction of various stages involved in direct multiple shoot regeneration from *in vitro* roots is presented in Fig 1. For induction of direct multiple shoot, the whole *in vitro* root explants (Fig 1A) were excised and cultured in MSN medium containing various cytokinins (ADS, BAP, KIN, and ZEA) and GA$_3$. Surprisingly, the multiple shoot proliferation (100% response) was observed only in two different PGR combinations, *viz*., MSN+BAP 1.0+GA$_3$ 1.0 mg L$^{-1}$ and MSN+BAP 1.0+GA$_3$ 2.0 mg L$^{-1}$ (Table 1; Fig 1B–1G). The intact root explants showed direct regeneration signs in both the media after 6–8 wks of inoculation. However, no response was observed in the other cytokinin (ADS, KIN and ZEA) supplemented media even upto 90 d of inoculation. In the process of direct regeneration, nodule-like structures appeared on the

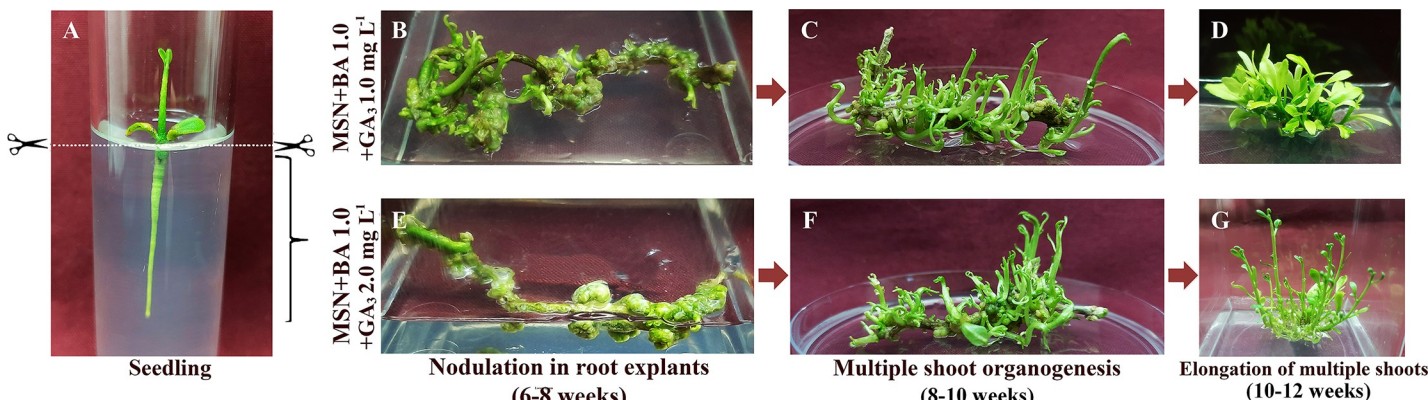

**Fig 1.  A-G** Direct multiple shoot organogenesis from *in vitro* root explants of *Citrus jambhiri* Lush. [**A**: Seedling; **B**, **C**, **D**: Nodulation in root explants, multiple shoot organogenesis, multiple shoot elongation at MSN+BA 1.0+GA$_3$ 1.0 mg L$^{-1}$, respectively; **E**, **F**, **G**: Nodulation in root explants, multiple shoot organogenesis, multiple shoot elongation at MSN+BA 1.0+GA$_3$ 2.0 mg L$^{-1}$, respectively]. DOI 10.17605/OSF.IO/CFT75.

**Table 1. Effect of Nitsch vitamin, cytokinins and gibberellic acid (GA$_3$) on direct multiple shoots organogenesis from *in vitro* root explants of *Citrus jambhiri* Lush.**

| Concentrations (mg L$^{-1}$) | | | Explant response (%) | Days to shoot initiation | Shoots/explant | Shoot length (cm) | Leaves/explant |
|---|---|---|---|---|---|---|---|
| MSN | Control | | NR | 0.0±0.0 a (0.71±0.0) | 0.0±0.0 a (0.71±0.0) | 0.0±0.0 a (0.71±0.0) | 0.0±0.0 a (0.71±0.0) |
| | ADS 1.0 | GA$_3$ 1.0 | NR | 0.0±0.0 a (0.71±0.0) | 0.0±0.0 a (0.71±0.0) | 0.0±0.0 a (0.71±0.0) | 0.0±0.0 a (0.71±0.0) |
| | | GA$_3$ 2.0 | NR | 0.0±0.0 a (0.71±0.0) | 0.0±0.0 a (0.71±0.0) | 0.0±0.0 a (0.71±0.0) | 0.0±0.0 a (0.71±0.0) |
| | BAP 1.0 | GA$_3$ 1.0 | 100 | 50.9±2.7 b (7.17±0.19) | 34.3±4.1 c (5.89±0.36) | 2.4±0.07 b (1.69±0.02) | 44.4±4.0 c (6.70±0.30) |
| | | GA$_3$ 2.0 | 100 | 75.6±4.1 c (8.72±0.23) | 26.8±2.5 b (5.22±0.25) | 2.5±0.05 c (1.73±0.01) | 36.5±2.4 b (6.08±0.20) |
| | KIN 1.0 | GA$_3$ 1.0 | NR | 0.0±0.0 a (0.71±0.0) | 0.0±0.0 a (0.71±0.0) | 0.0±0.0 a (0.71±0.0) | 0.0±0.0 a (0.71±0.0) |
| | | GA$_3$ 2.0 | NR | 0.0±0.0 a (0.71±0.0) | 0.0±0.0 a (0.71±0.0) | 0.0±0.0 a (0.71±0.0) | 0.0±0.0 a (0.71±0.0) |
| | ZEA 1.0 | GA$_3$ 1.0 | NR | 0.0±0.0 a (0.71±0.0) | 0.0±0.0 a (0.71±0.0) | 0.0±0.0 a (0.71±0.0) | 0.0±0.0 a (0.71±0.0) |
| | | GA$_3$ 2.0 | NR | 0.0±0.0 a (0.71±0.0) | 0.0±0.0 a (0.71±0.0) | 0.0±0.0 a (0.71±0.0) | 0.0±0.0 a (0.71±0.0) |
| | | SEm | - | 0.04 | 0.06 | 0.003 | 0.05 |
| | | CD (0.05) | - | 0.12 | 0.17 | 0.010 | 0.14 |
| | | CD (0.01) | - | 0.16 | 0.23 | 0.013 | 0.19 |

NR: No response; Data represent means ± standard deviation. Mean values in the parentheses are indicative of square root transformation. Data represents mean of two experiments with five replications each. Mean values followed by different letters in the same column indicate significant difference according to *Tukey's* test ($P \leq 0.01$). DOI 10.17605/OSF.IO/F9U35.

entire length of the roots (Fig 1B and 1E) after 6–8 wks of culture initiation, which turned green and produced shoot buds. Multiple shoot induction was achieved in both the media (Fig 1C and 1F), which grew to shootlets by 10–12 wks (Fig 1D and 1G). The medium containing GA$_3$ 1.0 mg L$^{-1}$ induced earlier bud break (50.9 d) by 25 d than that with GA$_3$ 2.0 mg L$^{-1}$ (75.6 d). Interestingly, MSN+BAP 1.0+GA$_3$ 1.0 mg L$^{-1}$ exhibited a significantly higher number of shoots (34.3) per explant with 44.4 numbers of leaves in 10–12 wks. An increase in GA$_3$ concentration in the media (MSN+BAP 1.0+GA$_3$ 2.0 mg L$^{-1}$) resulted in 26.8 shoots and 36.5 leaves per explant. The mean shoot length was comparatively greater (2.5 cm) in the medium with GA$_3$ 2.0 mg L$^{-1}$ than that with GA$_3$ 1.0 mg L$^{-1}$ (2.4 cm) [Table 1].

## Effect of auxins on root induction

The regenerated shoots were transferred to the rooting medium supplemented with various auxins like NAA and IAA. Fig 2 represents the response of ½MSN and MSN media with different concentrations of NAA and IAA on root induction. The responses of rooting, in terms of percent rooting, days to root initiation, root numbers per explant, and root length (cm) were significantly higher in ½MSN medium (Fig 2A) while compared with MSN (Table 2). Rooting responses ranged between 30–90% in ½MSN medium and 30–80% in MSN medium. ½MSN resulted in better rooting response (90%), while poor rooting (40%) was observed in full MSN. IAA (Fig 2D and 2E) induced significantly higher rooting among the auxins than NAA (Fig 2B and 2C) in both the culture media. Early rooting (9.4 d) was observed in the shootlets cultured on ½MSN+IAA (1.0 mg L$^{-1}$). Moreover, ½MSN+IAA (1.0 mg L$^{-1}$) medium induced a higher number of roots (3.5) per explant with an average root length of 4.4 cm within 9.4 d followed by the auxins free ½MSN medium, which induced 3.1 roots having 4.1 cm of mean root length in 9.7 d. The overall rooting response decreased at a higher concentration of NAA and IAA (1.5 and 2.0 mg L$^{-1}$) in ½MSN medium. On the contrary, rooting was better in MSN medium supplemented with auxins over control (Fig 2F–2J). In MSN medium, IAA (1.5–2.0 mg L$^{-1}$) resulted in more number of roots (1.8) with longer root length (2.6–3.1 cm) as compared to NAA and auxin free MSN (Table 2).

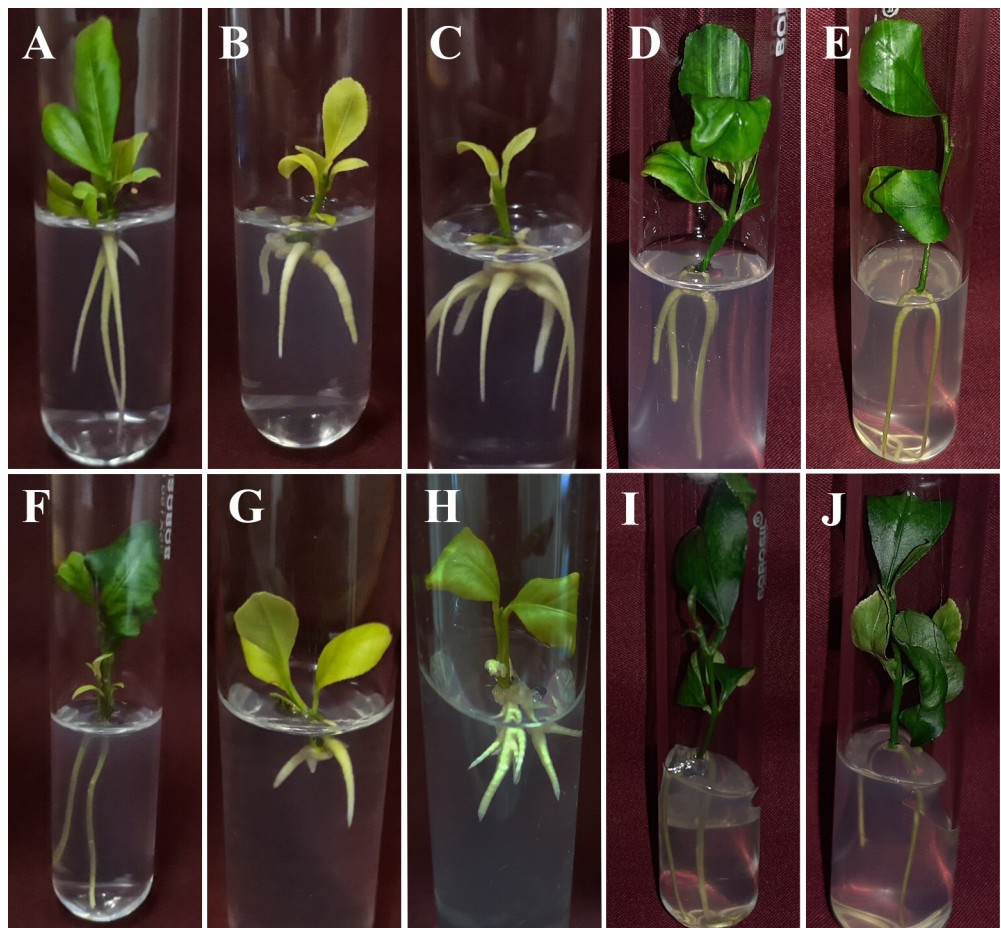

**Fig 2. A-J** Root induction in *in vitro* plantlets of *Citrus jambhiri* Lush. [**A**: ½MSN; **B-C**: ½MSN+NAA 1.0 and 2.0 mg L$^{-1}$; **D-E**: ½MSN+IAA 1.0 and 2.0 mg L$^{-1}$; **F**: MSN; **G-H**: MSN+NAA 1.0 and 2.0 mg L$^{-1}$; **I-J**: MSN+IAA 1.0 and 2.0 mg L$^{-1}$]. DOI 10.17605/OSF.IO/FR7X5.

## Gene expression analyses

The relative expression of growth regulating factors (*GRF1* and *GRF5*) and GA$_3$ pathway genes (*GA2OX1* and *KOI*) showed significant variations in different stages (stage I to stage IV) of direct multiple shoot induction in MSN media supplemented with BAP and GA$_3$. Expression of *GRF1* and *GRF5* genes was higher at all the stages in MSN+BAP 1.0+GA$_3$ 1.0 mg L$^{-1}$ compared to the same with GA$_3$ 2.0 mg L$^{-1}$ (Fig 3A and 3B). However, the expression of *GRF5* was higher than *GRF1* across all the stages. The *GA2OX1* gene showed a similar trend of significantly higher expression in MSN+BAP+GA$_3$ (1.0 mg L$^{-1}$, each) throughout all the stages (Fig 3C), whereas the expression of the *KO1* gene was higher in MSN+BAP 1.0+GA$_3$ 2.0 mg L$^{-1}$ (Fig 3D).

Fig 4 depicts auxin regulating genes expression in the roots derived from different auxin supplemented media. Both the auxin response factors (*ARF1* and *ARF8*) were expressed significantly at a higher level at ½ MSN+IAA 1.0 mg L$^{-1}$ than the same at ½ MSN+NAA 1.0 mg L$^{-1}$ (Fig 4A and 4B). Similarly, auxin efflux carrier protein genes (*PIN1* and *PIN5*), auxin/IAA family protein gene (*IAA4*), and *GH3* family protein gene (*GH3*) had higher level of expression in the roots of IAA supplemented media than NAA ones (Fig 4C to 4F). The mRNA transcripts of all the genes were relatively more abundant towards the later stage of root development (8 wks) than at 4 wks of culture.

**Table 2. Effect of Nitsch vitamin and auxins on root induction in the shootlets derived from *in vitro* root explants of *Citrus jambhiri* Lush.**

| | Concentrations (mg L$^{-1}$) | Rooting response (%) | Days to root initiation | Roots/Explant | Root length (cm) |
|---|---|---|---|---|---|
| ½MSN | ½MSN (control) | 90 | 9.7±4.5 a | 3.1±0.7 f | 4.1±0.8 g |
| | | | (3.14±0.65) | (1.89±0.18) | (2.14±0.20) |
| | NAA 0.5 | 40 | 14.2±9.0 b | 1.3±1.4 b | 2.7±3.0 ef |
| | | | (2.54±1.78) | (1.25±0.55) | (1.60±0.91) |
| | NAA 1.0 | 60 | 14.0±10.9 de | 2.5±1.7 e | 3.2±2.2 f |
| | | | (3.03±1.77) | (1.65±0.58) | (1.82±0.69) |
| | NAA 1.5 | 40 | 11.3±8.8 bc | 1.6±1.8 c | 1.7±1.9 d |
| | | | (2.24±1.68) | (1.33±0.65) | (1.36±0.67) |
| | NAA 2.0 | 30 | 25.0±14.0 h | 1.0±1.2 a | 1.1±1.7 ab |
| | | | (2.44±2.39) | (1.18±0.50) | (1.15±0.64) |
| | IAA 0.5 | 60 | 9.9±5.3 ab | 1.1±1.0 ab | 2.4±1.5 e |
| | | | (2.69±1.20) | (1.22±0.39) | (1.62±0.55) |
| | IAA 1.0 | 80 | 9.4±4.3 a | 3.5±1.4 g | 4.4±1.8 g |
| | | | (3.08±0.73) | (1.97±0.37) | (2.19±0.41) |
| | IAA 1.5 | 70 | 10.6±3.8 b | 1.6±0.7 c | 2.6±1.2 e |
| | | | (3.29±0.59) | (1.43±0.27) | (1.74±0.31) |
| | IAA 2.0 | 60 | 16.0±10.5 f | 1.8±1.4 cd | 1.4±1.2 c |
| | | | (3.30±1.73) | (1.44±0.52) | (1.31±0.47) |
| MSN | MSN (control) | 40 | 13.3±9.2 d | 1.1±1.1 ab | 1.3±1.4 c |
| | | | (2.44±1.78) | (1.19±0.46) | (1.25±0.54) |
| | NAA 0.5 | 30 | 13.3±7.3 d | 1.0±0.9 a | 0.9±0.9 a |
| | | | (2.51±1.65) | (1.16±0.42) | (1.14±0.41) |
| | NAA 1.0 | 80 | 27.3±12.9 i | 1.9±1.2 d | 2.6±1.5 e |
| | | | (4.34±2.07) | (1.49±0.48) | (1.68±0.55) |
| | NAA 1.5 | 40 | 22.2±15.3 g | 1.3±1.4 b | 1.3±1.7 bc |
| | | | (3.06±2.36) | (1.25±0.55) | (1.22±0.60) |
| | NAA 2.0 | 40 | 29.8±16.7 j | 1.2±1.7 b | 1.6±2.2 cd |
| | | | (2.62±2.60) | (1.17±0.64) | (1.28±0.78) |
| | IAA 0.5 | 30 | 10.5±5.8 a | 0.9±0.9 a | 0.9±0.8 a |
| | | | (2.27±1.43) | (1.13±0.40) | (1.11±0.37) |
| | IAA 1.0 | 30 | 13.5±8.0 d | 1.3±2.0 b | 1.2±1.9 b |
| | | | (1.90±1.69) | (1.19±0.70) | (1.14±0.67) |
| | IAA 1.5 | 80 | 11.7±4.7 c | 1.8±0.3 cd | 3.1±1.0 f |
| | | | (3.44±0.69) | (1.51±0.09) | (1.87±0.27) |
| | IAA 2.0 | 70 | 11.1±6.5 bc | 1.8±1.3 cd | 2.6±1.8 e |
| | | | (2.82±1.33) | (1.45±0.48) | (1.66±0.61) |
| | SEm | - | 0.73 | 0.21 | 0.25 |
| | CD (0.05) | - | 2.06 | 0.59 | 0.7 |
| | CD (0.01) | - | 2.74 | 0.79 | 0.93 |

Data represent means ± standard deviation. Mean values in the parentheses are indicative of square root transformation. Data represents mean of two experiments with five replications each. Mean values followed by different letters in the same column indicate significant difference according to *Tukey's* test (*P*≤0.01). DOI 10.17605/OSF.IO/P7SFZ.

## Scanning electron microscopy (SEM)

The SEM illustration of the roots derived from ½MSN with different auxin concentrations (NAA and IAA) is represented in Fig 5. Morphologically, the roots obtained in auxin free

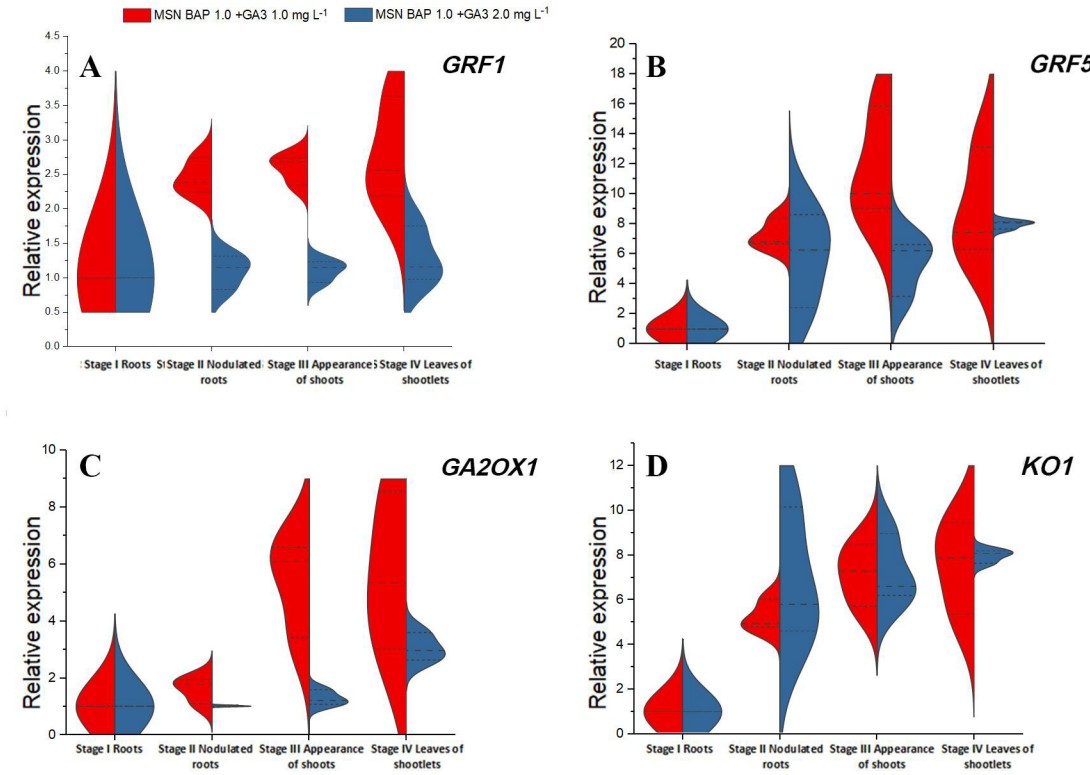

**Fig 3. A-D** Relative expression of growth regulating factors (**A**. *GRF1* and **B**. *GRF5)* and GA$_3$ pathway genes (**C**. *GA2OX1* and **D**. *KO1*) at different stages of organogenesis from *in vitro* root explants of *Citrus jambhiri* Lush. DOI 10.17605/OSF.IO/XGUFT.

½MSN and ½MSN+IAA 1.0 mg L$^{-1}$ were similar to those obtained in ½MSN+NAA 1.0 mg L$^{-1}$ (Fig 5A to 5C). External root micrographs showed better structure and texture when cultured in ½MSN+IAA 1.0 mg L$^{-1}$ (Fig 5C1 and 5C2). Concerning internal micrograph, better anatomy was observed in ½MSN+IAA 1.0 mg L$^{-1}$ (Fig 5C3 and 5C4), followed by auxin free ½MSN (Fig 5A3 and 5A4). Larger pore sizes in the cross-sections were observed in the roots derived from ½MSN+NAA 1.0 mg L$^{-1}$ (Fig 5B3 and 5B4).

## Indexing of the regenerants for HLB and CTV

Assessment of the mother plants, *in vitro* regenerants and *in vivo* infected plants (as positive controls) for HLB-bacterium and CTV was carried out using PCR based detection system. For detection of HLB, the *Las* primers, designed from hyper variable effector protein of *C*La, were used to give an amplicon of 500 bp in the positive control samples (Fig 6A). Similarly, the *K607* and *K608* primers was used to detect CTV, showing an amplicon of 648 bp (Fig 6B). However, no bands were observed at the respective amplicon size among the regenerants for both HLB bacterium and CTV, except in positive controls (Fig 6A and 6B). The results indicated that the plantlets obtained from root explants under various concentrations of PGRs are free of these two devastating pathogens.

## Assessment of genetic stability

Genetic fidelity of randomly selected *in vitro* regenerants and the mother plants maintained *in vivo* was assessed using RAPD and ISSR markers (Fig 7A and 7B). A Monomorphic profile

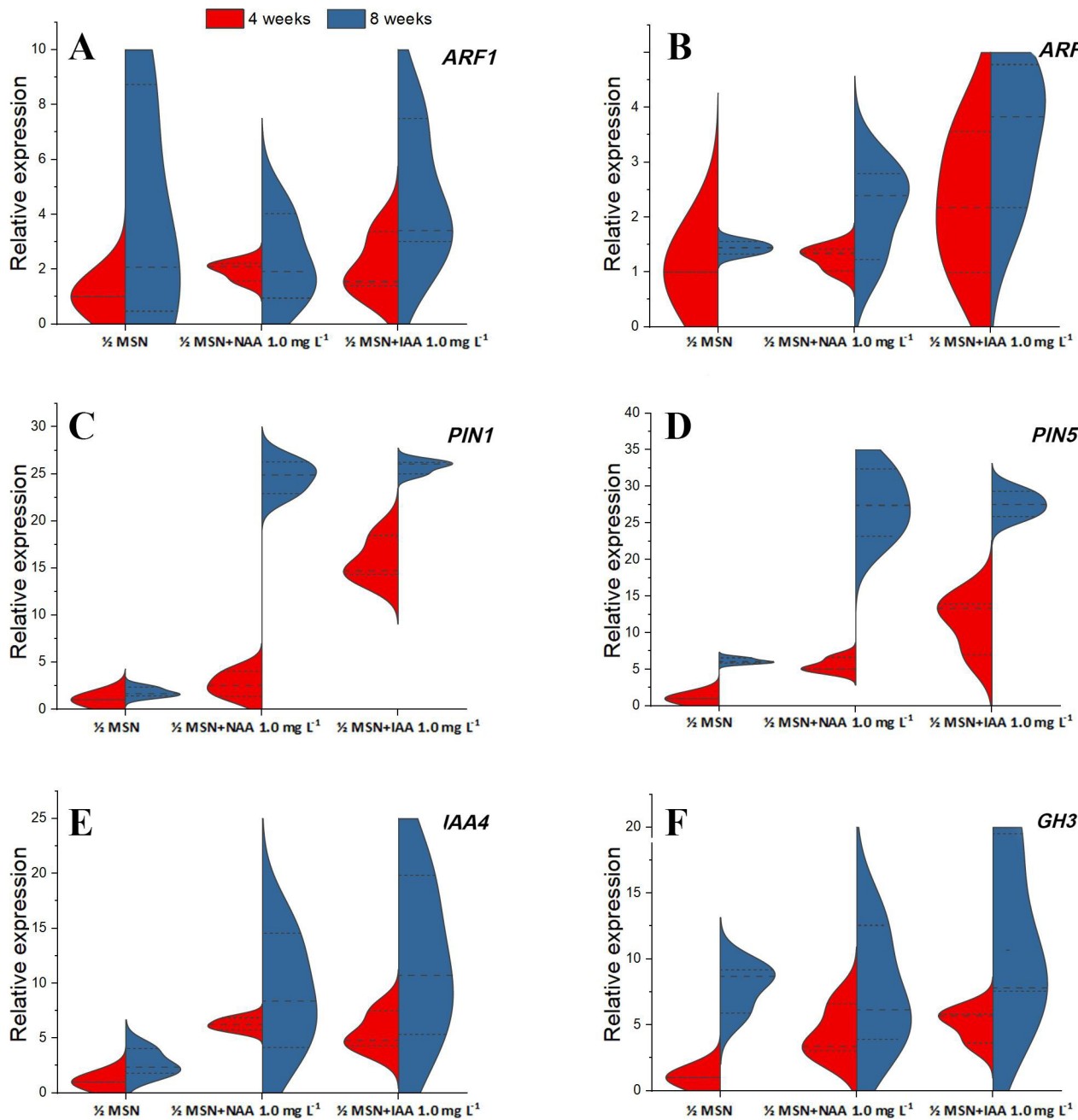

**Fig 4. A-F** Relative expression of auxin responsive factors (**A**. *ARF1* and **B**. *ARF8*), auxin efflux carrier protein genes (**C**. *PIN1* and **D**. *PIN5*), auxin/IAA family protein gene (**E**. *IAA4*) and GH3 family protein gene (**F**. *GH3*) at different stages of rooting of *Citrus jambhiri* Lush. DOI 10.17605/OSF. IO/C27QB.

was observed among the regenerants and the mother plant for all the sixteen primers (Table 3). Eight RAPD primers yielded 40 scorable monomorphic bands ranging from 100 bp to 2000 bp (Fig 7A and S1 Fig), whereas 46 bands were amplified by eight ISSR markers, between 250 bp to 5000 bp (Fig 7B and S2 Fig). The results of the present study authenticated the genetic homogeneity amongst the regenerants obtained from the root explants of *C. jambhiri* Lush.

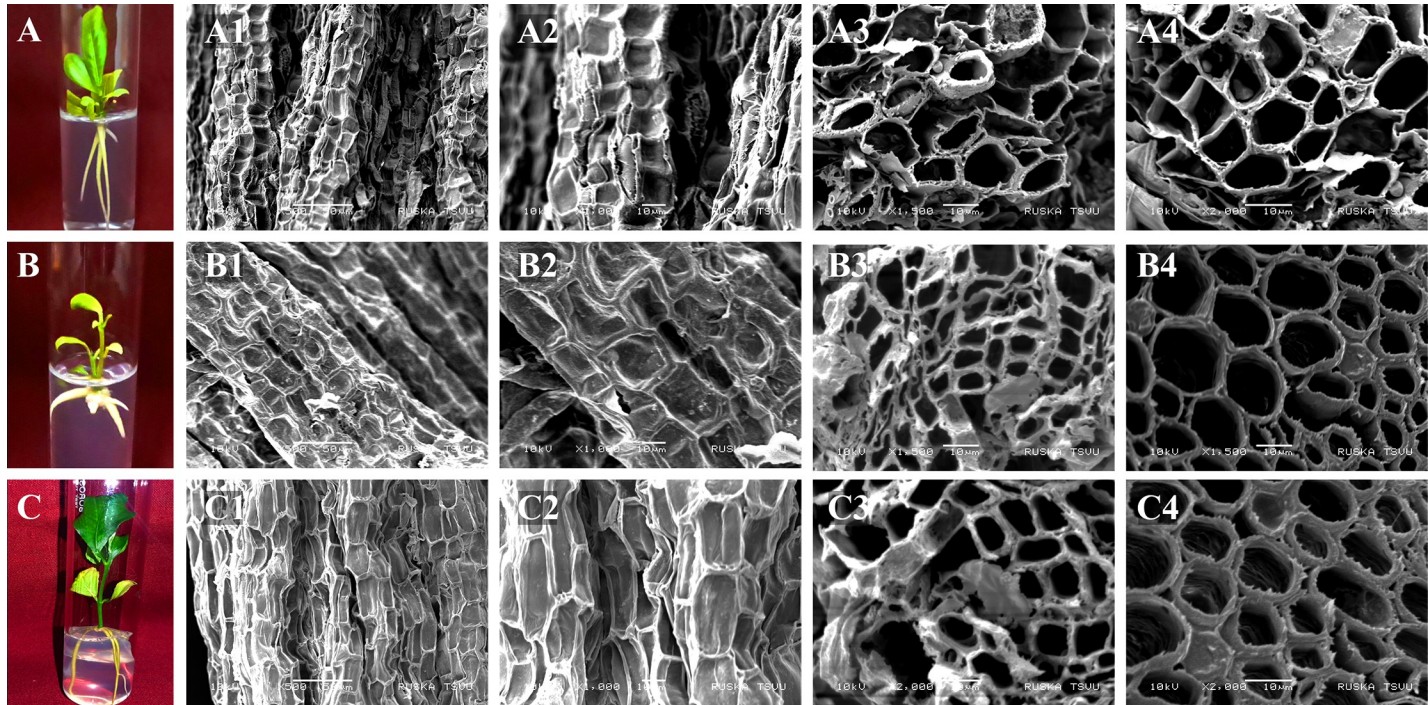

**Fig 5. A-C** Scanning electron micrograph of the roots induced in *Citrus jambhiri* Lush. plantlets using various concentrations of auxins [**A**. ½MSN (**A1-A2**: External at 500 and 1000x; **A3-A4**: Internal at 1500 and 2000x), **B**. ½MSN+NAA 1.0 mg L$^{-1}$(**B1-B2**: External at 500 and 1000x; **B3-B4**: Internal at 1500 and 2000x), **C**. ½MSN+IAA 1.0 mg L$^{-1}$ (**C1-C2**: External at 500 and 1000x; **C3-C4**: Internal at 1500 and 2000x)]. DOI 10.17605/OSF.IO/GP6AH.

## Acclimatization and hardening

The well-grown plantlets with 2–4 leaves and roots were subsequently acclimatized, hardened off and transplanted in polyhouse after bacterium and virus indexing and genetic fidelity assessment (Fig 8A to 8F). In our study, the plantlets derived from MSN+BAP 1.0+GA$_3$ 1.0 mg L$^{-1}$ and rooted in auxin free ½MSN resulted in higher survivability (94%) *in vivo*, followed by ½MSN+IAA (0.5–1.0 mg L$^{-1}$, 90–92%) [Fig 8G].

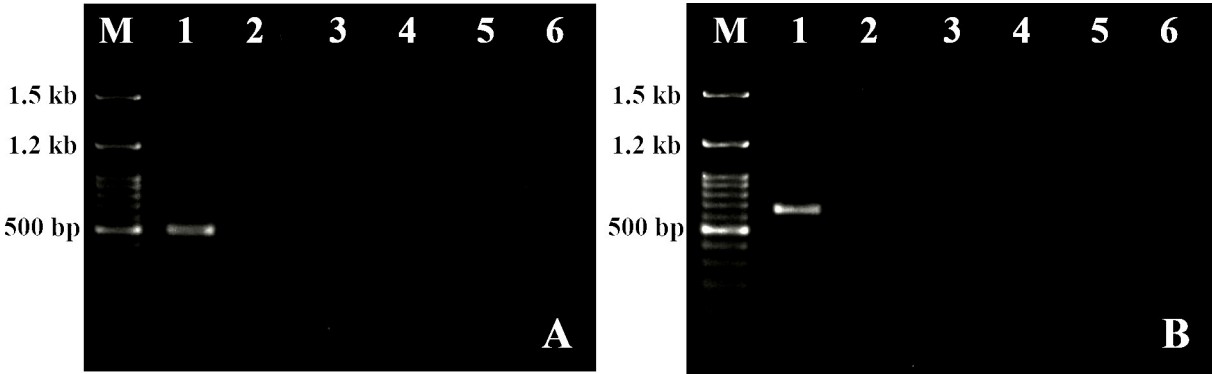

**Fig 6. A-B** Indexing of the regenerants employing *Las* F and R (500 bp) to detect Citrus greening (Huanglongbing) [**A**] and *K607* F and *K608* R (648 bp) to detect *Citrus tristeza* retro virus [**B**] in *Citrus jambhiri* Lush. regenerants [Lane M: 100 bp ladder (GCC biotech); Lane 1: Positive control (C. greening or CTV infected leaf midrib); Lane 2: leaf midrib from the mother plant; Lane 3: *In vitro* leaf midrib from the plantlets (½MSN); Lane 4: *In vitro* leaf midrib from the plantlets (½MSN+NAA 1.0 mg L$^{-1}$); Lane 5: *In vitro* leaf midrib from the plantlets (½MSN+IAA 1.0 mg L$^{-1}$); Lane 6: primer control (negative control)]. DOI 10.17605/OSF.IO/YDNKU.

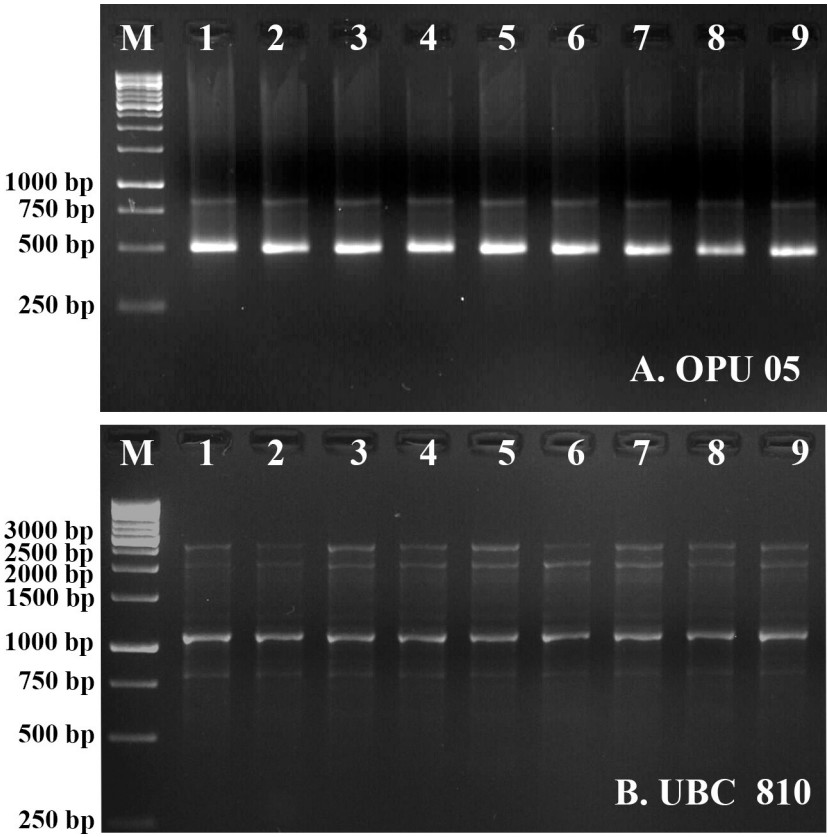

**Fig 7. A-B** Genetic fidelity of *Citrus jambhiri* Lush. regenerants employing RAPD (**A**. OPU 05) and ISSR (**B**. UBC 810) markers [Lane M: 1 kb DNA ladder for RAPD and ISSR; Lane 1: Mother plant from Kachai village, Ukhrul, Manipur; Lane 2: *In vitro* seedlings; Lane 3: Seedlings planted at Langol farm of ICAR, Manipur; Lane 4: Seedlings planted at polyhouse of ICAR, Manipur; Lane 5: Regenerants obtained from MSN+BAP 1.0+GA$_3$ 1.0 mg L$^{-1}$; Lane 6: Regenerants obtained from MSN+BAP 1.0+GA$_3$ 2.0 mg L$^{-1}$; Lane 7: Plantlets obtained from ½MSN; Lane 8: Plantlets obtained from ½MSN+NAA 1.0 mg L$^{-1}$; Lane 9: Plantlets obtained from ½MSN+IAA 1.0 mg L$^{-1}$]. DOI 10.17605/OSF.IO/9YC6V.

## Discussion

This study emphasized *in vitro* clonal propagation *via* direct shoot organogenesis from whole root explants. The key success in achieving direct organogenesis depends on the explant type and media composition, which influence the regeneration pathways [37]. Therefore, the optimization of media compositions for specific explants is essential to establish an efficient regeneration protocol. In *Citrus jambhiri*, root segments (1 cm) from 4 wks old seedlings used for direct regeneration in MS media supplemented with various doses of BAP (0.5–3.0 mg L$^{-1}$), resulted in 52% shoot bud induction [9]. Basal MS media is the most preferred one in *C. jambhiri* regeneration to date [2, 5, 6]; however, no report is available on the effect of Nitsch vitamins on shoot organogenesis from the whole *in vitro* roots. Cytokinins and auxins play a major role in the growth of *in vitro* shoots in the culture media, and the balance between these two decides the fate of organogenesis. Cytokinins, particularly BAP, were reported to induce shoot buds in almost all species of *Citrus* [38]; however, the optimal concentration may vary from species to species. In the present study, different levels of ADS, BAP, KIN, and ZEA in combination with GA$_3$ were tested for direct organogenesis (Table 1). Of the various cytokinins tested, the only response in terms of shoot bud induction was achieved with BA 1.0 mg L$^-$

**Table 3. Polymerase chain reaction amplicons obtained from RAPD and ISSR analysis of *in vitro* regenerants of *C. jambhiri* Lush.**

|  | Sl. No. | Primer ID | Primer sequence (5'-3') | Total bands amplified | Number of monomorphic bands | Band size (bp) |
|---|---|---|---|---|---|---|
| RAPD | 1 | OPA 09 | 5' GGGTAACGCC 3' | 5 | 5 | 400–1000 |
|  | 2 | OPC 01 | 5' TTCGAGCCAG 3' | 5 | 5 | 300–2000 |
|  | 3 | OPC 08 | 5' TGGACCGGTG 3' | 4 | 4 | 400–1200 |
|  | 4 | OPC 12 | 5' TGTCATCCCC 3' | 5 | 5 | 400–2000 |
|  | 5 | OPD 02 | 5' GGACCCAACC 3' | 8 | 8 | 300–1200 |
|  | 6 | OPF 02 | 5' GAGGATCCCT 3' | 6 | 6 | 150–1500 |
|  | 7 | OPU 05 | 5' TTGGCGGCCT 3' | 2 | 2 | 500–800 |
|  | 8 | OPU 20 | 5' ACAGCCCCCA 3' | 5 | 5 | 100–700 |
|  |  | **Total bands** |  | **40** | **40** |  |
| ISSR | 9 | UBC-807 | (AG)8T | 7 | 7 | 450–5000 |
|  | 10 | UBC-810 | (GA)8T | 9 | 9 | 800–2500 |
|  | 11 | UBC-811 | (GA)8C | 7 | 7 | 250–2000 |
|  | 12 | UBC-812 | (GA)8A | 3 | 3 | 1000–2000 |
|  | 13 | UBC-827 | (AC)8G | 4 | 4 | 800–2500 |
|  | 14 | UBC-840 | (GA)8YT | 5 | 5 | 1300–2200 |
|  | 15 | UBC-855 | (AC)8YT | 7 | 7 | 400–1800 |
|  | 16 | UBC-880 | (GGGTG)3 | 4 | 4 | 500–1300 |
|  |  | **Total bands** |  | **46** | **46** |  |

DOI 10.17605/OSF.IO/RXQHS.

[1], which confirms the role of BAP on direct shoot bud induction [9]. Among the two doses of GA$_3$ tested along with BAP, the media MSN+BAP 1.0+GA$_3$ 1.0 mg L$^{-1}$ induced early bud break with a higher number of shoots and leaves from the *in vitro* root explants compared to

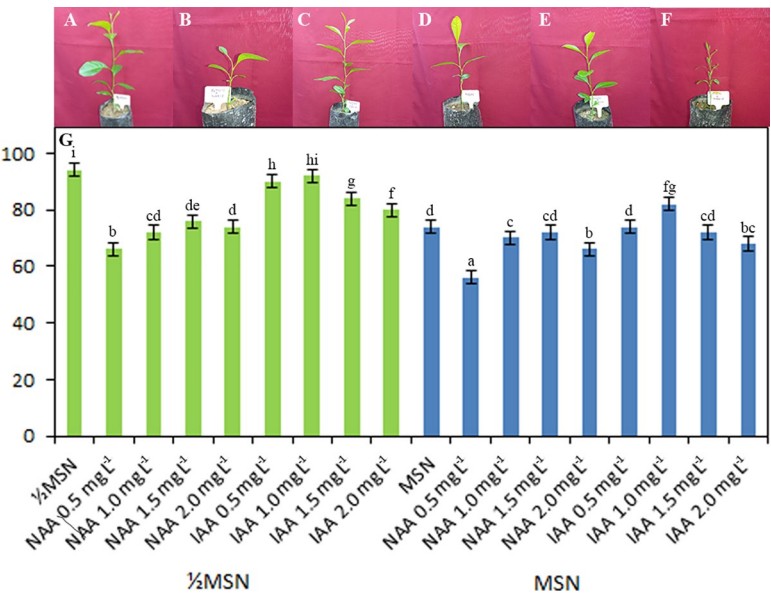

**Fig 8. A-G** Survivability of *in vitro* plantlets from different concentrations of auxins [**A**. ½MSN, **B**. ½MSN+NAA (1.5 mg L$^{-1}$), **C**. ½MSN+IAA (1.0 mg L$^{-1}$), **D**. MSN, **E**. MSN+NAA (1.5 mg L$^{-1}$), **F**. MSN+IAA (1.0 mg L$^{-1}$), **G**. % Survivability of the *in vitro* plantlets derived from MSN+BA 1.0+GA$_3$ 1.0 mg L$^{-1}$ and different concentrations of auxins] DOI 10.17605/OSF.IO/R67JZ.

MSN+BAP 1.0+GA$_3$2.0 mg L$^{-1}$. The higher dose of BAP (3.0 mg L$^{-1}$) induced shoots from cotyledon explants of *C. jambhiri* with only 13% response [10]. However, in our study, a lower dose of BAP (1.0 mg L$^{-1}$) in combination with GA$_3$ (1.0–2.0 mg L$^{-1}$) induced multiple shoots from *in vitro* root explants (100%). The lower concentrations of BAP might have a stimulating effect on shoot initiation by reducing the dominance of newly formed shoot buds [9]. GA$_3$ has a significant influence on shoot elongation [39]; thus, media combination supplemented with GA$_3$ 2.0 mg L$^{-1}$ resulted in higher shoot length. In addition to BAP and GA$_3$, the media contained the Nitsch vitamins rich in folic acid (0.5 mg L$^{-1}$) and d-biotin (0.05 mg L$^{-1}$). The addition of folic acid and d-biotin might have enriched the medium and the higher concentration of other constituents like nicotinic acid (5 mg L$^{-1}$), glycine (2 mg L$^{-1}$), pyridoxine HCl (0.5 mg L$^{-1}$) and thiamine HCl (0.5 mg L$^{-1}$) might have become congenial for organogenesis from whole root explants efficiently. Folic acid or folates are also known as B9 vitamins and play an essential role in carbon and nitrogen metabolism in plants and control signaling cascades [40]. This regeneration system became highly efficient as 34.3 numbers of shoots (Fig 1C and 1F) were regenerated from a single root explant within 10–12 wks of culture.

The establishment of a profuse root system ensures the efficacy of an efficient regeneration protocol. Distinctive root distribution in *in vitro* regenerants influences nutrient uptake efficiency in *Citrus* plants *in vivo*. Auxins like NAA, IBA, and IAA are mostly useful to induce roots in *in vitro* plantlets. Different concentrations of NAA (0.5–10.0 mg L$^{-1}$) and IBA (1.5–2.0 mg L$^{-1}$) in MS medium have been used successfully to induce roots in different *Citrus* species [1, 10, 41]. However, an efficient root induction in the auxin free basal media (½ strength of MSN) was achieved in the present study. Auxin free ½MSN evidently induced longer roots (4.1 cm) with better penetration ability (Fig 2A). Among the auxins, NAA (0.5–1.0 mg L$^{-1}$) resulted in short and bulky roots attaining the length of 3.2 cm with retarded shoot growth and etiolated leaves (Fig 2B and 2C). Further increase in NAA concentrations delays rooting and reduces the number of roots and root length. On the contrary, IAA (0.5–1.0 mg L$^{-1}$) induced early rooting (9.4 d) with a better root system (3.5 number of roots and 4.4 cm in length) morphologically similar to the roots obtained in auxin free ½MSN (Fig 2D and 2E). IAA at lower doses (0.5–1.0 mg L$^{-1}$) may be incorporated into ½MSN for better nutrient uptake efficiency and establishment of the plantlets in the soil as evident from the results.

To understand the mechanisms underlying growth responses owing to various PGR combinations, a comparative gene expression analysis was performed for growth regulating factors (*GRF*s) at different stages of direct shoot initiation. In the present study, level of expressions of *GRF1* and *GRF5* was significantly high under MSN+BAP 1.0+GA$_3$ 1.0 mg L$^{-1}$, which was positively correlated with the nodulation, shoot bud initiation and proliferation from the *in vitro* root explants. The expressions of both the *GRF*s were more in young leaves than roots, which confirms the previous findings [23]. However, the expression level of *GRF5* was significantly high than that of *GRF1* during all the developmental stages (Fig 3A to 3B). GRFs govern organ growth by managing cell proliferation [42]. The presence of *cis*-elements (GARE and P-BOX) in the promoter region of *GRF*s is associated with GA-response and transcription activation [43]. The promoter of *GRF1* contains one GARE element, whereas that of *GRF5* contains one GARE and one P-box component [23]. In this study, the higher level of expression of *GRF5* could be attributed to the presence of two GA-responsive *cis*-elements.

The GA$_3$ biosynthesis genes are reported to have a significant role in regulating plant growth and development [39]. Being the modulator of plant growth and development, gibberellins can promote shoot and leaf elongation. In the present study, the expression of *GA2OX1* and *KO1* was studied in various shoot regeneration stages. The level of *GA2OX1* expression was up to a 5-fold increase in leaf tissues than that in roots (Fig 3C). Moreover, a relatively high level of expression of *KO1* was observed in root and leaf tissues in media MSN+BAP 1.0

+GA$_3$ 2.0 mg L$^{-1}$ than the same in GA$_3$1.0 mg L$^{-1}$ (Fig 3D). Higher level of expression of *KO1* might be due to higher GA$_3$ concentration in the media used [30].

Various auxin-responsive genes (*ARF1*, *ARF8*, *PIN1*, *PIN5*, *IAA4*, and *GH3*) were analysed in *in vitro* roots to understand the effect of exogenous auxins like NAA and IAA. The results of qRT-PCR revealed a significantly higher expression level of all auxin-responsive genes in roots obtained in IAA supplemented media (Fig 4). The greater abundance of mRNAs of *ARF1* and *ARF8* (4-fold), *IAA4*, and *GH3* (11-fold), followed by *PIN1* and *PIN5* (25-fold) was observed in the roots growing in IAA media, implicating the role of IAA in auxin regulation in these roots. The role of these genes in the induction of adventitious roots is well explained [30, 44, 45].

This study revealed that the auxin free ½MSN medium induced an efficient rooting system as valuable to the medium supplemented with auxin. Auxin medium, particularly, ½MSN +NAA, induced a callus at the base of the shootlets initially and subsequently differentiated into bulky roots. The same medium resulted in stunted shoot growth (Fig 5B) probably due to the low nutrient uptake ability, attributed to the largely porous root structure (Fig 5B3 and 5B4) as examined through SEM. SEM analysis is a well-proven tool to visualize the vascular architect of the roots. Appropriate root compaction with small or medium porosity allows smooth uptake of nutrients and water through xylem and phloem when examined using SEM [29], which was evident in this study. Results of SEM analysis also revealed that ½MSN+IAA resulted in the proportionate alignment of cells and tissues in the root with better root porosity (Fig 5C3 and 5C4), appropriate to the functioning of vascular system for healthy and vigorous growth of the plantlet. Therefore, incorporation of IAA (1.0 mg L$^{-1}$) into the medium could lead to better shootlets development.

HLB, caused by *Candidatus* Liberibacter asiaticus- a fastidious, vector transmitted bacterium and CTV- a retrovirus, are the most economically important diseases, reside accross the midrib of the leaves, affecting the majority of the *Citrus* plants [21]. *In vitro* regeneration process limits the extent of vector transmitted diseases from mother plant-explant-regenerants. Moreover, RT-PCR-based detection and confirmation of viruses are well documented in *Citrus* [46]. In the present study, by using RT-PCR, no amplicons were detected for HLB bacterium and CTV in *in vitro* regenerants (Fig 6) obtained from the root explants, which authenticated the quality of plantlets as free from HLB and CTV.

The main aim of *in vitro* propagation techniques is to obtain true-to-type plantlets without genetic variations compared to the mother plants [47]. Somaclonal variations are generally introduced among the regenerants during the process of *in vitro* culture. Besides, there could be variations among the regenerants, which might be due to the *'artifact'* of *in vitro* techniques, which probably occurred due to the PGR concentrations at shoot proliferation and root induction stages. Assessment of genetic homogeneity employing multiple molecular markers authenticates the genetic stability of micro-propagated plants [48]. Among the molecular markers, RAPD is the potent PCR-based DNA markers used to investigate genetic homogeneity among the *in vitro* regenerants and the mother plants of *Citrus jambhiri* [1, 2]. On the other hand, ISSR is an efficient multilocus DNA marker for genetic polymorphism studies with higher PIC (polymorphism information content) values and better reproducibility than RAPD [49]. No genetic variations were observed among the regenerants and the mother plant in the present study while examining their genetic homogeneity using RAPD and ISSR markers (Fig 7). All amplified bands (40 and 46 for RAPD and ISSR, respectively) were found to be monomorphic, establishing the clonal fidelity of the regenerants. Apparently, the present study is the first report on ISSR marker-assisted genetic fidelity study in *in vitro* regenerants of *Citrus jambhiri*.

The success and efficiency of *in vitro* regeneration technique depend on the successful establishment of true-to-type *in vitro* plantlets in soil [1]. Earlier reports showed a maximum

83% survivability of *in vitro* regenerated *Citrus jambhiri* plantlets [3]. Following this protocol, 94% plantlets (Fig 8G) obtained through MSN+BAP 1.0+GA$_3$ 1.0 mg L$^{-1}$ and rooted in ½MSN (auxin-free) were successfully established under field conditions, followed by 90–92% of the plantlets derived through ½MSN+IAA (0.5–1.0 mg L$^{-1}$) with better growth and development.

## Conclusions

In conclusion, we have derived a simple two-step efficient protocol from *in vitro* whole root explants of *Citrus jambhiri* Lush. in MSN medium. The first step included direct profuse multiple shoot organogenesis from *in vitro* root explants in MSN+BAP 1.0+GA$_3$ 1.0 mg L$^{-1}$, which produced over 34 regenerants per explant. The second step involved root induction in the regenerants using ½ strength of MSN that produced more than three roots per explant in less than ten days with better root structures examined through SEM. Direct regeneration from *in vitro* root explants did not induce somaclonal variations in the regenerants compared to the mother plant. The level of expression of *GRF* and GA$_3$ regulating genes were higher throughout the stages of shoot development, whereas the profiling of auxin regulating genes reflected higher expression in *in vitro* roots while grown under ½MSN+IAA (0.5–1.0 mg L$^{-1}$). The regenerants were tested fastidious bacteria- and virus-free, as evident from the indexing studies. The plantlets showed 94% survivability *in vivo*. This is the first report on direct high-efficient shoot regeneration from the *in vitro* root explant of *Citrus jambhiri* Lush.

## Supporting information

**S1 Fig. Polymerase chain reaction amplicons obtained from RAPD primers (OPA 09, OPC 01, OPC 08, OPC 12, OPD 02, OPF 02 OPU 05 and OPU 20) analysis of *in vitro* regenerants of *C. jambhiri* Lush.** [Lane M: 1 kb DNA ladder for RAPD; Lane 1: Mother plant from Kachai village, Ukhrul, Manipur; Lane 2: *In vitro* seedlings; Lane 3: Seedlings planted at Langol farm of ICAR, Manipur; Lane 4: Seedlings planted at polyhouse of ICAR, Manipur; Lane 5: Regenerants obtained from MSN+BAP 1.0+GA$_3$ 1.0 mg L$^{-1}$; Lane 6: Regenerants obtained from MSN+BAP 1.0+GA$_3$ 2.0 mg L$^{-1}$; Lane 7: Plantlets obtained from ½MSN; Lane 8: Plantlets obtained from ½MSN+NAA 1.0 mg L$^{-1}$; Lane 9: Plantlets obtained from ½MSN+IAA 1.0 mg L$^{-1}$]. DOI 10.17605/OSF.IO/XWT8B.
(TIF)

**S2 Fig. Polymerase chain reaction amplicons obtained from ISSR primers (UBC-807, UBC-810, UBC-811, UBC-812, UBC-827, UBC-840, UBC-855 and UBC-880) analysis of *in vitro* regenerants of *C. jambhiri* Lush.** [Lane M: 1 kb DNA ladder for ISSR; Lane 1: Mother plant from Kachai village, Ukhrul, Manipur; Lane 2: *In vitro* seedlings; Lane 3: Seedlings planted at Langol farm of ICAR, Manipur; Lane 4: Seedlings planted at polyhouse of ICAR, Manipur; Lane 5: Regenerants obtained from MSN+BAP 1.0+GA$_3$ 1.0 mg L$^{-1}$; Lane 6: Regenerants obtained from MSN+BAP 1.0+GA$_3$ 2.0 mg L$^{-1}$; Lane 7: Plantlets obtained from ½MSN; Lane 8: Plantlets obtained from ½MSN+NAA 1.0 mg L$^{-1}$; Lane 9: Plantlets obtained from ½MSN+IAA 1.0 mg L$^{-1}$]. DOI 10.17605/OSF.IO/JBZUX.
(TIF)

**S1 Raw images. RAPD: DOI 10.17605/OSF.IO/AZ9CS.**
(PDF)

**S2 Raw images. ISSR: DOI 10.17605/OSF.IO/B2XPR.**
(PDF)

**S1 Table. List of quantitative real-time PCR (qRT-PCR) primers used for gene expression analyses of *Citrus jambhiri* Lush. regenerants.**
(DOCX)

**S2 Table. List of RAPD and ISSR primers used for genetic fidelity studies of *Citrus jambhiri* Lush. regenerants.**
(DOCX)

## Acknowledgments

The authors acknowledge the support of the RUSKA Laboratory, Hyderabad, India, for skilful assistance with the SEM analyses and Dr. S. K. Sharma, ICAR RC NEHR, Manipur Centre for help with virus indexing experiments. The infrastructure facility provided by the Director and Joint Director, Indian Council of Agricultural Research (ICAR) Research Complex for NEH Region, Manipur Centre, India, is gratefully acknowledged. Academic support by Visva-Bharati-A Central University to T. Roshni Devi for pursuing her Ph. D. is duly acknowledged.

## Author Contributions

**Conceptualization:** Madhumita Dasgupta, Manas Ranjan Sahoo.

**Data curation:** Madhumita Dasgupta.

**Formal analysis:** Tongbram Roshni Devi, Madhumita Dasgupta, Manas Ranjan Sahoo.

**Funding acquisition:** Manas Ranjan Sahoo.

**Investigation:** Tongbram Roshni Devi, Madhumita Dasgupta.

**Methodology:** Tongbram Roshni Devi.

**Project administration:** Paresh Chandra Kole, Narendra Prakash.

**Supervision:** Manas Ranjan Sahoo, Paresh Chandra Kole.

**Validation:** Madhumita Dasgupta, Paresh Chandra Kole.

**Writing – original draft:** Tongbram Roshni Devi, Madhumita Dasgupta, Manas Ranjan Sahoo.

**Writing – review & editing:** Madhumita Dasgupta, Manas Ranjan Sahoo, Paresh Chandra Kole.

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
