## [Decision Letter · Decision Letter 0]

14 Dec 2020

PONE-D-20-35054

High efficient de novo root-to-shoot organogenesis in Citrus jambhiri Lush.: gene expression, genetic stability and virus indexing

PLOS ONE

Dear Dr. SAHOO,

Thank you for submitting your manuscript to PLOS ONE. After careful consideration, we feel that it has merit but does not fully meet PLOS ONE’s publication criteria as it currently stands. Therefore, we invite you to submit a revised version of the manuscript that addresses the points raised during the review process.

We look forward to receiving your revised manuscript.

Kind regards,

Jen-Tsung Chen, Ph.D.

Academic Editor

PLOS ONE

Journal Requirements:

2. In your Methods section, please provide additional location information, including geographic coordinates for the data set if available.

6. Please include captions for your Supporting Information files at the end of your manuscript, and update any in-text citations to match accordingly. Please see our Supporting Information guidelines for more information: http://journals.plos.org/plosone/s/supporting-information

Reviewers' comments:

Reviewer's Responses to Questions

**Comments to the Author**

1. Is the manuscript technically sound, and do the data support the conclusions?

Reviewer #1: Partly

Reviewer #2: Yes

Reviewer #3: Yes

2. Has the statistical analysis been performed appropriately and rigorously? 

Reviewer #1: Yes

Reviewer #2: Yes

Reviewer #3: Yes

3. Have the authors made all data underlying the findings in their manuscript fully available?

Reviewer #1: No

Reviewer #2: Yes

Reviewer #3: Yes

4. Is the manuscript presented in an intelligible fashion and written in standard English?

Reviewer #1: No

Reviewer #2: Yes

Reviewer #3: Yes

5. Review Comments to the Author

Reviewer #1: The manuscript entitled “High efficient de novo root-to-shoot organogenesis in Citrus jambhiri Lush.: gene expression, genetic stability, and virus indexing” is nicely planned and executed. Though it can be considered for possible publication in Plos One, the current version still needs improvement. Please address the below-mentioned comments for further improvement.

Line 28, 45, please mention the common name on the first appearance in the abstract and the main text.

Line 37, please follow the standard unit style throughout the text, i.e., mg L-1.

Line 86, define RAPD and ISSR. Why authors choose RAPD markers along with ISSR? Although the experiment is well planned and executed, however, I got some reservations concerning RAPD markers that are known to have low reproducibility and thus can impact the reliability of results. If the experiment is replicated to ensure the PCR results were reproducible, then there are no issues with the accuracy of results given in this manuscript.

Line 92, define GA.

Line 93, please avoid starting the sentence with an abbreviation (GA).

Line 219-222, please mention which statistical tool/software was used for the data analysis.

Please provide a sharper image for fig 6.

In section “Assessment of genetic stability”, please provide the complete data sets in terms of remaining gel images or arrange all the generated data in a table (suppl. table/figures). Currently, there are only two gel images for two primers only. How about the results of other markers? Therefore, please provide us with the other data as a suppl. material.

Please be consistent with analyse or anlyze. Also, check the entire text for grammar and spacing errors. The English language needs improvement.

Reviewer #2: In this paper the authors showed the “High efficient de novo root-to-shoot organogenesis in Citrus jambhiri Lush.: gene expression, genetic stability and virus indexing”. The authors have studied some parameters related to phenomenon of root-to-shoot organogenesis. This is a well written article and standard for such a high-ranking journal. Overall, it is a good presented paper. Language quality is need be improved. I could recommend it for publication

Reviewer #3: My comments:

- please add the novelty and aims to ABSTRACT.

-INTRODUCTION is so long, please revise it.

-All figures related to gene expression are not professionally drawn, so I suggest to be redrawn them with a suitable software.

6. PLOS authors have the option to publish the peer review history of their article (what does this mean?). If published, this will include your full peer review and any attached files.

Reviewer #1: No

Reviewer #2: **Yes: **mona Soliman

Reviewer #3: No

---

## [Author Response · Author response to Decision Letter 0]

25 Jan 2021

Attached as 'Response to the reviewers'

---

## [Decision Letter · Decision Letter 1]

29 Jan 2021

High efficient de novo root-to-shoot organogenesis in Citrus jambhiri Lush.: gene expression, genetic stability and virus indexing

PONE-D-20-35054R1

Dear Dr. SAHOO,

We’re pleased to inform you that your manuscript has been judged scientifically suitable for publication and will be formally accepted for publication once it meets all outstanding technical requirements.

Kind regards,

Jen-Tsung Chen, Ph.D.

Academic Editor

PLOS ONE

Additional Editor Comments (optional):

Reviewers' comments:

Reviewer's Responses to Questions

**Comments to the Author**

1. If the authors have adequately addressed your comments raised in a previous round of review and you feel that this manuscript is now acceptable for publication, you may indicate that here to bypass the “Comments to the Author” section, enter your conflict of interest statement in the “Confidential to Editor” section, and submit your "Accept" recommendation.

Reviewer #1: All comments have been addressed

2. Is the manuscript technically sound, and do the data support the conclusions?

Reviewer #1: Yes

3. Has the statistical analysis been performed appropriately and rigorously? 

Reviewer #1: Yes

4. Have the authors made all data underlying the findings in their manuscript fully available?

Reviewer #1: Yes

5. Is the manuscript presented in an intelligible fashion and written in standard English?

Reviewer #1: Yes

6. Review Comments to the Author

Reviewer #1: Dear Authors,

Thank you for revising the MS according to the proposed comments and suggestions. Notably, the MS quality has been improved, and it should be considered for acceptance. Therefore, I am endorsing the MS for acceptance. Cheers!

7. PLOS authors have the option to publish the peer review history of their article (what does this mean?). If published, this will include your full peer review and any attached files.

Reviewer #1: No

---

## [Editor Report · Acceptance letter]

9 Feb 2021

PONE-D-20-35054R1 

High efficient *de novo* root-to-shoot organogenesis in *Citrus jambhiri* Lush.: gene expression, genetic stability and virus indexing 

Dear Dr. Sahoo:

I'm pleased to inform you that your manuscript has been deemed suitable for publication in PLOS ONE. Congratulations! Your manuscript is now with our production department. 

Kind regards, 

on behalf of

Dr. Jen-Tsung Chen 

Academic Editor

PLOS ONE